# Enhancing Medical Image Segmentation via Heat Conduction Equation

**Rong Wu** 🆔                                          RONG.WU@UCSF.EDU
**Yim-Sang Yu**                                    YIMSANG.YU@UCSF.EDU
*Department of Epidemiology and Biostatistics, University of California, San Francisco, CA, USA*

## Abstract

Medical image segmentation models struggle to achieve efficient global context modeling and long-range dependency reasoning under practical computational budgets. In this work, we propose a hybrid architecture utilizing U-Mamba with Heat Conduction Equation, which combines state-space modules for efficient long-range reasoning with Heat Conduction Operators (HCOs) in the bottleneck layers, simulating frequency-domain thermal diffusion for enhanced semantic abstraction. Experimental results show that our model attains the highest DSC (0.8719) on the Abdomen CT dataset. It suggests that blending state-space dynamics with heat-based global diffusion offers a scalable solution for medical segmentation tasks.

**Keywords:** Medical image segmentation, U-Net, Heat conduction, State space model

## 1. Introduction

Emerging techniques in image representation, inspired by physical processes such as Heat Conduction and frequency-domain transformations (Rao et al., 2021; Wang et al., 2025), are moving beyond the standard spatial domain and showing great promise. These advancements highlight the significant potential of integrating physical principles with deep learning to further improve medical image segmentation. To capitalize on this, we introduce a hybrid architecture named U-Mamba-HCO (UMH). This network integrates the U-Mamba encoder with physics-inspired HCOs placed strategically in the bottleneck. The proposed UMH network is designed to simultaneously address two critical challenges in biomedical image analysis: capturing long-range dependencies and facilitating global semantic diffusion, all while maintaining high computational efficiency. This hybridization is effective because it seamlessly combines two complementary properties: (1) Long-Range Dependency Modeling: Inherited from Mamba's efficient and selective structured state-space mechanism, enabling the network to process extensive contextual information, (2) Low Computational Complexity: Inspired by vHeat (Wang et al., 2025), we adapt the HCOs to effectively enhance global feature extraction with low computational complexity and high interpretability.

## 2. Method

**Preliminaries.** In this paper, we proposed to extend the heat equation to 3D region $D \in \mathbb{R}^3$ as following: $\frac{\partial u}{\partial t} = k(\frac{\partial^2 u}{\partial x^2} + \frac{\partial^2 u}{\partial y^2} + \frac{\partial^2 u}{\partial z^2})$, And the proposed general solution of heat equation should be: $u(x, y, z, t) = \mathcal{F}^{-1}(\tilde{f}(\omega_x, \omega_y, \omega_z)e^{-k(\omega_x^2 + \omega_y^2 + \omega_z^2)t})$. If we consider the

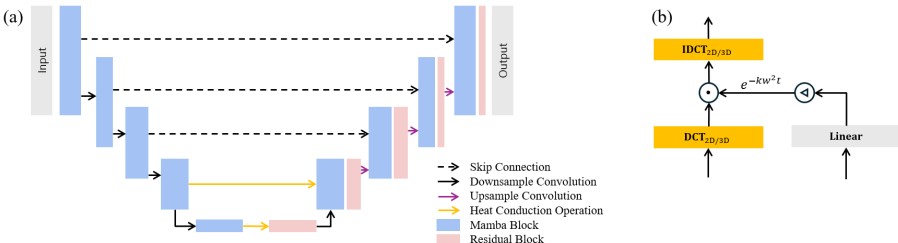

Figure 1: Overall pipeline. (a) Encoder–Decoder design. (b) DCT/IDCT applies an adaptive thermal diffusion filter, simulating heat conduction.

input 3D images as $U(x, y, z, c, 0)$ and outputs as $U(x, y, z, c, t)$, where $c$ is the number of channel, then $U^t = \mathcal{F}^{-1}[\mathcal{F}(U^0)e^{-k(\omega_x^2+\omega_y^2+\omega_z^2)t}]$. By utilizing the discrete version of the (inverse) Fourier Transform, we replace the (inverse) Fourier Transform with the discrete cosine transformation. The detailed derivation is provided in Appendix A. As is shown in Fig. 1(b), the implementation can be expressed as: $U^t = \mathbf{IDCT_{2D/3D}}[\mathbf{DCT_{2D/3D}}(U^0)e^{-k\sum_i \omega_i^2 t}]$, where $i = \{x, y\}$ in 2D and $i = \{x, y, z\}$ in 3D are the dimension, and $k$ denotes the thermal diffusivity, which is estimated from the extracted features within the frequency domain.

**Overview** Figure 1(a) illustrates the overall structure of our proposed U-Mamba-HCO (UMH) network. The architecture builds upon the U-Mamba Encoder–Decoder framework (Ma et al., 2024a) and introduces HCO layers (Wang et al., 2025) at the two bottleneck regions to enhance global semantic diffusion and long-range dependency modeling. This design integrates the selective state-space reasoning capability of Mamba with the physics-inspired frequency-domain diffusion of HCO, achieving an efficient yet interpretable representation learning mechanism for biomedical image segmentation. Specifically, The upper encoder stages consist of standard U-Mamba blocks, each containing two residual convolutional blocks followed by a Mamba module. At the bottleneck, we replace the two skip connections with HCO layers, which perform frequency-domain global filtering via DCT/IDCT. Each HCO models feature propagation as thermal diffusion, where image patches act as heat sources and the diffusion coefficient is adaptively predicted via learnable frequency value embeddings (FVEs) (Wang et al., 2025). This substitution enables interpretable, content-adaptive global interaction at the network's semantic core with computational complexity of only $\mathcal{O}(N^{1.5})$. The code is available at `https://github.com/Rows21/UMH`.

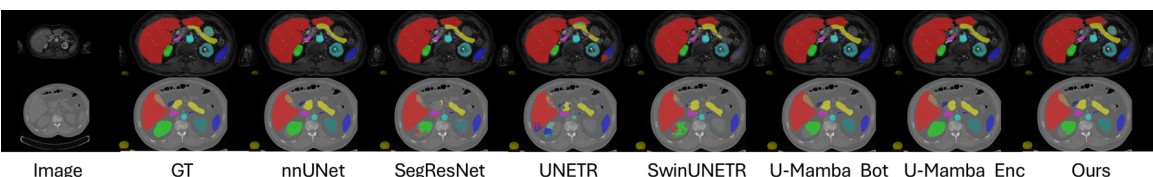

Figure 2: Semantic segmentation results for UMH and its competitors.

## 3. Experiments

**Dataset, Evaluation Metrics and Implementation Details.** To evaluate our proposed method across different segmentation targets, and modalities, we apply our algorithm on **Abdomen CT** and **Abdomen MRI** dataset. The detail information is described in Appendix B. We compared UMH with nnUNet (Isensee et al., 2021), SegResNet (Myronenko, 2018), UNETR (Hatamizadeh et al., 2022), SwinUNETR (Hatamizadeh et al., 2021), and U-Mamba (Ma et al., 2024a). We used the default image preprocessing in nnUNet (Isensee et al., 2021).

Table 1: Comparison with SOTA methods on 3D segmentation tasks.

| Methods | Abdomen CT (3D) | | Abdomen MRI (3D) | |
|---|---|---|---|---|
| | DSC (%)↑ | NSD (%)↑ | DSC (%)↑ | NSD (%)↑ |
| nnUNet | 86.15±7.90 | 89.72±8.24 | 83.09±7.69 | 89.96±7.29 |
| SegResNet | 79.27±11.62 | 82.57±11.94 | 81.46±9.59 | 88.41±9.17 |
| UNETR | 68.24±15.06 | 70.04±15.77 | 68.67±14.88 | 74.40±16.27 |
| SwinUNETR | 75.94±10.95 | 76.63±11.90 | 75.65±13.94 | 82.18±14.09 |
| U-Mamba_Bot | 86.83±8.08 | **90.49±8.21** | 84.53±6.73 | 91.21±6.34 |
| U-Mamba_Enc | 86.38±9.08 | 89.80±9.21 | **85.01±7.32** | **91.71±6.89** |
| UMH | **87.19±6.28** | 90.37±7.16 | 84.84±6.85 | 91.53±6.51 |

Note: ↑ means higher is better, the best and second-best values are in **bold** and ___.

**Main Results.** We evaluate UMH on 3D abdominal organ segmentation tasks, with results summarized in Table 1 and Figure 2. Table 1 shows that across all datasets, UMH achieves either the best or second-best performance in 3D segmentation tasks in both DSC and NSD metrics. UMH attains the highest DSC (0.8719) and NSD (0.9037) on the Abdomen CT dataset, outperforming all competing methods, UMH ranks second yet remains highly competitive with U-Mamba_Enc in Abdomen MRI task, demonstrating strong generalization across imaging modalities. Figure 2 visualizes the qualitative results, highlighting the merits of UMH over competing methods. While most methods suffer from target incompleteness (false negatives) and background misclassification (false positives), UMH consistently produces correct boundaries that are demonstrably more consistent with the ground truth. Ablation studies are also conducted with results in Appendix C.

## 4. Conclusions

This paper introduces UMH, a novel hybrid framework that combines state-space modeling and physics-inspired diffusion. UMH integrates Mamba modules into the encoder for efficient long-range feature learning and inserts HCOs into the bottleneck to simulate frequency-domain thermal diffusion, enabling global semantic abstraction with linear complexity. Experiments on 3D abdominal CT/MRI datasets show that UMH consistently outperforms strong baselines achieving the highest Dice and NSD scores in most settings.

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

## Appendix A. Preliminaries

Let $u(x, y, t)$ denote the temperature of point $(x, y)$ at time $t$ within a two-dimensional region $D \in \mathbb{R}^2$, the classic physical heat equation can be performed as:

$$\frac{\partial u}{\partial t} = k\left(\frac{\partial^2 u}{\partial x^2} + \frac{\partial^2 u}{\partial y^2}\right), \tag{1}$$

where $k > 0$ is the thermal diffusivity constant, measuring the rate of heat transfer in a material. By setting the initial condition $u(x, y, t)|_{t=0}$ to $f(x, y)$, the general solution of Eq. (2) can be derived by applying the Fourier Transform (FT, denoted as $\mathcal{F}$) to both sides of the equation, which gives:

$$\mathcal{F}\left(\frac{\partial u}{\partial t}\right) = k\mathcal{F}\left(\frac{\partial^2 u}{\partial x^2} + \frac{\partial^2 u}{\partial y^2}\right). \tag{2}$$

Denoting $\tilde{u}(\omega_x, \omega_y, t)$ as the FT-transformed form of $u(x, y, t)$, i.e., $\tilde{u}(\omega_x, \omega_y, t) := \mathcal{F}(u(x, y, t))$, the left-hand-side of Eq. (2) can be written as:

$$\mathcal{F}\left(\frac{\partial u}{\partial t}\right) = \frac{\partial \tilde{u}(\omega_x, \omega_y, t)}{\partial t}, \tag{3}$$

and by leveraging the derivative property of FT, the righthand-side of Eq. (2) can be transformed as

$$\mathcal{F}\left(\frac{\partial^2 u}{\partial x^2} + \frac{\partial^2 u}{\partial y^2}\right) = -(\omega_x^2 + \omega_y^2)\tilde{u}(\omega_x, \omega_y, t). \tag{4}$$

Therefore, by combining the expression of both sides of the equation, Eq. (2) can be formulated as an ordinary differential equation (ODE) in the frequency domain, which can be written as:

$$\frac{d\tilde{u}(\omega_x, \omega_y, t)}{dt} = -k(\omega_x^2 + \omega_y^2)\tilde{u}(\omega_x, \omega_y, t) \tag{5}$$

By setting the initial condition $\tilde{u}(\omega_x, \omega_y, t)|_{t=0}$ to $\tilde{f}(\omega_x, \omega_y)$, which denotes the FT-transformed $f(x, y)$), $\tilde{u}(\omega_x, \omega_y, t)$ in Eq. (6) can be solved as

$$\tilde{u}(\omega_x, \omega_y, t) = \tilde{f}(\omega_x, \omega_y)e^{-k(\omega_x^2 + \omega_y^2)t}. \tag{6}$$

Finally, the general solution of heat equation in the spatial domain can be obtained by performing inverse Fourier Transformer ($\mathcal{F}^{-1}$) on Eq. (7), which gives the following expression:

$$u(x, y, t) = \mathcal{F}^{-1}(\tilde{f}(\omega_x, \omega_y)e^{-k(\omega_x^2 + \omega_y^2)t}). \tag{7}$$

In this paper, we proposed to extend the heat equation to three-dimensional region $D \in \mathbb{R}^3$ as following:

$$\frac{\partial u}{\partial t} = k\left(\frac{\partial^2 u}{\partial x^2} + \frac{\partial^2 u}{\partial y^2} + \frac{\partial^2 u}{\partial z^2}\right), \tag{8}$$

And the proposed general solution of heat equation should be:

$$u(x, y, z, t) = \mathcal{F}^{-1}(\tilde{f}(\omega_x, \omega_y, \omega_z)e^{-k(\omega_x^2 + \omega_y^2 + \omega_z^2)t}). \tag{9}$$

If we consider the input 3D images as $U(x, y, z, c, 0)$ and outputs as $U(x, y, z, c, t)$, where $c$ is the number of channel, then $U^t = \mathcal{F}^{-1}[\mathcal{F}(U^0)e^{-k(\omega_x^2 + \omega_y^2 + \omega_z^2)t}]$. By utilizing the discrete version of the (inverse) Fourier Transform, we replace the (inverse) Fourier Transform with the discrete cosine transformation. As is shown in Fig. 1(b), the implementation can be expressed as:

$$U^t = \mathbf{IDCT_{2D}}[\ \mathbf{DCT_{2D}}(U^0)e^{-k(\omega_x^2 + \omega_y^2)t}], \tag{10}$$

$$U^t = \mathbf{IDCT_{3D}}[\ \mathbf{DCT_{3D}}(U^0)e^{-k(\omega_x^2 + \omega_y^2 + \omega_z^2)t}], \tag{11}$$

where parameter $k$ denotes the thermal diffusivity, which is estimated from the extracted features within the frequency domain.

## Appendix B. Datasets and Implementation Details

To evaluate our proposed method across different segmentation targets, and modalities, we apply our algorithm on **Abdomen CT** and **Abdomen MRI** dataset, which are adopted and modified from MICCAI 2022 FLARE Challenge (Ma et al., 2024b) and AMOS Challenge (Ji et al., 2022). Both of them focused on the segmentation of 13 abdominal organs. **Abdomen CT:** The training set contained 50 CT scans that were from the MSD Pancreas dataset (Simpson et al., 2019) and the annotations were from AbdomenCT-1K. Another 50 cases from different medical centers (Clark et al., 2013) were used for evaluation and the annotations were provided by the challenge organizers. **Abdomen MRI:** We used 60 labeled MRI scans for model training and 50 annotated MRI scans as the testing set, this dataset is released from U-Mamba (Ma et al., 2024a). Table 2 details the UMH network architecture used for each dataset. All the networks were trained from scratch for 1000 epochs on one NVIDIA A100 GPU. We used Dice Similarity Coefficient (DSC) and Normalized Surface Distance (NSD) for semantic segmentation tasks of organ segmentation in CT and MRI scans. The experimental setup involved a random 4:1 split for the training and validation subsets.

Table 2: Configurations for each dataset.

| Configurations | Patch Size | Stages | Pooling |
|---|---|---|---|
| CT (3D) | $(2, 40, 224, 192)$ | 6 | $(3, 3, 5)$ |
| MR (3D) | $(2, 48, 160, 224)$ | 6 | $(3, 5, 5)$ |

## Appendix C. Ablation Studies

Table 3 presents the ablation results of our proposed UMH framework on the 3D Abdomen CT segmentation task. We systematically evaluate the contributions of different architectural components, including the baseline **nnUNet** (Isensee et al., 2021), the encoder-integrated **U-Mamba_Enc**, the bottleneck-enhanced **U-HCO_Bot**, and the encoder-level

**U-HCO_Enc**. While integrating the Mamba module into the encoder yields a moderate improvement in both DSC and NSD compared to the nnUNet baseline, the introduction of the HCO alone provides limited benefits when applied either at the bottleneck or encoder stage. However, when both components are jointly incorporated, the network achieves the best performance, outperforming all variants. This demonstrates that the HCO and Mamba modules are complementary: the Mamba layer enhances global context modeling through efficient sequence propagation, whereas the HCO layer improves local feature diffusion and boundary consistency. Their integration allows UMH to better capture both long-range dependencies and smooth spatial transitions, leading to more accurate and consistent organ delineation.

Table 3: Ablation study of UMH on 3D Abdomen CT task.

| Networks | DSC↑ | NSD↑ |
|---|---|---|
| nnUNet | $0.8615 \pm 0.0790$ | $0.8972 \pm 0.0824$ |
| U-Mamba_Enc | $0.8638 \pm 0.0908$ | $0.8980 \pm 0.0921$ |
| U-HCO_Bot | $0.8618 \pm 0.0941$ | $0.8965 \pm 0.0978$ |
| U-HCO_Enc | $0.8575 \pm 0.0854$ | $0.8895 \pm 0.0857$ |
| UMH | $\mathbf{0.8719 \pm 0.0628}$ | $\mathbf{0.9037 \pm 0.0716}$ |

