# OpenReview forum: "Enhancing Medical Image Segmentation via Heat Conduction Equation"
_MIDL.io/2026/Short_Papers — MIDL 2026 - Short Papers Poster_

### Official Review · Reviewer_wRAo · 2026-04-24
**Interesting Method, weak results**

**Rating:** 3
**Confidence:** 4

**Review:**

The paper proposes combining U-Mamba and HCO components in a new UNet-type network for supervised segmentation. While it is definitely interesting to highlight new architectures the claim of superiority of this type of model even for just the tasks/dataset presented is very weakly supported. If we only consider the mean of the scores, the box-standard nnUNet is only being outperformed by 2% in the two reported metrics. From the appendix we can however see that the standard deviation across the spread is in the range of 7%. The experiments themselves are appropriate, and present an interesting modification of the established models.

The contradiction of the annotation of the best scoring models and the actual values in Table 1 should be addressed.

**Summary:**

The authors propose a new U-Net architecture for supervised segmentation, by combining two known components (U-Mamba and HCO) into U-Mamba-HCO. This is then evaluate on two different segmentation datasets with Abdominal CT and MRI respectively. The comparison involves a few other well known baselines, like nnUNet, SegResNet, UNETR etc).

**Strengths:**

The paper is clearly written. The authors take some time to explain Heat Conducting Operators (HCO) again as it is not very well known yet compared to the other components. The comparisons include a good range of baselines for supervised segmentation.

**Weaknesses:**

- The UMH Model is reported to achieve the best results against all comparisons in the Abdominal CT experiments. This however contradicts Table 1 where the highest score for NSD (0.9049) is labelled as second highest, and the score for UMH (0.9037) is labelled as highest, even though it is lower.

- The performance increase compared to a standard nnUNet is within less than 2%, and measures of spread are missing. (Spread reported in the Ablation in the Appendix.)

- The authors emphasize the computational efficiency, but we have no information comparing any notion of efficiency between the different models used in the experiments.

**Justification Of Rating:**

There is weak evidence of the proposed model outperforming the state of the art. There are issues with the claims, but the reported performance is close enough to the SOTA for the novelity of the proposed model to merit consideration.

---

### Decision · Program_Chairs · 2026-05-08

Accept (Poster)